# Study of the Cross-Influence between III-V and IV Elements Deposited in the Same MOVPE Growth Chamber

**DOI:** 10.3390/ma14051066

**Published:** 2021-02-25

**Authors:** Gianluca Timò, Marco Calicchio, Giovanni Abagnale, Nicola Armani, Elisabetta Achilli, Marina Cornelli, Filippo Annoni, Bernd Schineller, Lucio Claudio Andreani

**Affiliations:** 1RSE, Strada Torre della Razza, le Mose, 29100 Piacenza, Italy; marco.calicchio@rse-web.it (M.C.); giovanni.abagnale@rse-web.it (G.A.); nicola.armani@rse-web.it (N.A.); elisabetta.achilli@rse-web.it (E.A.); marina.cornelli@rse-web.it (M.C.); filippo.annoni@rse-web.it (F.A.); 2Department of Physics, University of Pavia, Via Bassi 6, 27100 Pavia, Italy; lucio.andreani@unipv.it; 3AIXTRON SE, Dornkaulstrasse 2, 52134 Herzogenrath, Germany; B.Schineller@aixtron.com

**Keywords:** Ge, SiGe, SiGeSn, InGaP, InGaAs, AlGaAs, MOVPE, cross-doping, multi-junction solar cells

## Abstract

We have deposited Ge, SiGe, SiGeSn, AlAs, GaAs, InGaP and InGaAs based structures in the same metalorganic vapor phase epitaxy (MOVPE) growth chamber, in order to study the effect of the cross influence between groups IV and III-V elements on the growth rate, background doping and morphology. It is shown that by adopting an innovative design of the MOVPE growth chamber and proper growth condition, the IV elements growth rate penalization due to As “carry over” can be eliminated and the background doping level in both IV and III-V semiconductors can be drastically reduced. In the temperature range 748–888 K, Ge and SiGe morphologies do not degrade when the semiconductors are grown in a III-V-contaminated MOVPE growth chamber. Critical morphology aspects have been identified for SiGeSn and III-Vs, when the MOVPE deposition takes place, respectively, in a As or Sn-contaminated MOVPE growth chamber. III-Vs morphologies are influenced by substrate type and orientation. The results are promising in view of the monolithic integration of group-IV with III-V compounds in multi-junction solar cells.

## 1. Introduction

There is considerable interest in exploiting band gap engineering possibilities offered by the integration of III-V and IV-based materials for optoelectronic and more specifically for photovoltaic applications [1,2,3]. A successful integration of InGaP and Si has already been accomplished in the realization of dual junction solar cells by mechanical staking the different semiconductors [4]. More recently, triple junction InGaP/InGaAs/Si solar cells have been manufactured by applying the direct wafer bonding technique [5].

However, a cheaper and, in principle, a simpler solution for the integration of III-V and IV-based materials would be the manufacturing of monolithic architectures obtained by depositing III-V and group IV-based semiconductors inside the same metal organic vapor phase epitaxy (MOVPE) growth chamber. So far, this approach has been hindered by the cross influence between groups IV and III-V elements.

This cross influence can be manifested, for example, as a limitation in the growth rate. In case of GeSn deposition, it has been shown that arsenic (As) “carry-over” brings saturation of the surface sites, blocking the growth [6]. Additionally, R. Jakomin observed a strong change in the epitaxial germanium MOVPE growth regime as a function of the III-V elements concentration in the layers [7]. Moreover, one of the most important aspects of the cross influence between groups IV and III-V elements is the cross doping, which makes it difficult to control the layer’s conductivity. E. Welser et al., for example, showed doping memory effects of Ge in III-V alloys when they are grown in the same MOVPE growth chamber [8].

There are, therefore, two main issues to tackle in order to advance the realization of III-V and group IV-based monolithic architectures in the same MOVPE growth apparatus: (i) to control the As “carry-over”, so that the growth of IV-based materials can be carried out without a growth rate penalization, and (ii) to adopt proper growth conditions in order to reduce cross doping, so that the conductivities of both III-V and IV-based semiconductors can be modulated. A further issue still to be investigated is whether the cross influence between groups IV and III-V elements can also affect the morphology of III-V and IV-based semiconductors.

In this contribution, we address the above-mentioned issues, using a modified MOVPE growth chamber and carrying out an in depth growth analysis. The results show that the integration of group IV with III-V compounds in the same MOVPE growth chamber is feasible, with promising perspectives for the realization of high efficiency, low cost, multi-junction solar cell structures.

## 2. Materials and Methods

The III-V and group IV-based semiconductor deposition has been carried out by means of a AIX 2800G4 MOVPE “planetary” system (from AIXTRON SE, Herzogenrath, Germany), whose growth chamber has been preliminarily modified in order to reduce the cross doping between groups IV and III-V elements. The reactor modifications have concerned: (i) the use of a special triple gas injector with a bigger dimeter than standard, for enhancing the precursor utilization efficiency and thus reducing the parasitic deposition on the reactor walls, (ii) the insertion of a quartz plate in the center of graphite susceptor, which stays at a substantially lower temperature than the susceptor and thus allows minimizing the deposition at the chamber center zone upstream of the leading wafer edge, (iii) the optimization of thermal decoupling between the ceiling and the top reactor cooling plate, in order to decrease the unwanted parasitic deposition on the ceiling due to condensation. For a better control of the deposition uniformity in the kinetic regime, the use of double gas foil rotation (DGFR) system has been adopted. This system uses a different mixture of N_2_/H_2_ gases injected in the center zone and in the edge zone underneath the graphite satellite where the wafer is set, allowing an in-situ control of the temperature profile across the wafer. In Figure 1, the image of the described MOVPE growth chamber is depicted.

The growth of AlAs, GaAs, InGaAs and InGaP has been carried out by using AsH_3_, PH_3_, TEGa (Ga(C_2_H_6_)_3_), TMGa (Ga(CH_3_)_3_), TMAl (Al(CH_3_)_3_) and TMIn (In(CH_3_)_3_) as main precursors, by DEZn (Zn(C_2_H_6_)_2_) as dopant, by using H_2_ as carrier gas. The deposition of Ge, SiGe and SiGeSn has been accomplished by using GeH_4_ (10% in H_2_ or 10% in N_2_), Si_2_H_6_ (1000 ppm in H_2_ or 10% in H_2_) and SnCl_4_, by using N_2_ as carrier gas [9].

The reactor pressure has been kept constant at 50 mbar, while the growth temperature has been varied between 743 K and 923 K.

Four-inch and six-inch Ge substrates with orientation (100) 6° off towards <111> have been used, as well as four-inch GaAs substrates with orientation (100), 2° off towards <110>.

The growth rate has been evaluated by fitting the MOVPE in-situ reflectivity curves (Laytec EpiCurve^®^ TT, Berlin, Germany) or by scanning electron microscope (SEM, Zeiss-sigma, Oberkochen, Germany), analysis in cross-section. Layer morphology has been evaluated by optical microscope (Leica DM 4000 M, Weizlar, Germany) with 500× magnification. Structural characterization has been carried out by high resolution X-ray diffraction (HRXRD) and Grazing Incidence X-Ray Diffraction (GIXRD) with a BRUKER D8 Discovery system (BRUKER, Ettlingen, Germany). The atoms profile has been measured by secondary ion-mass spectroscopy (SIMS) (Cameca ims 4f-E6, with Cs+ ions at 5.5 keV, Gennevilliers Cedex, France) while the carrier profiles have been extensively measured by electrochemical capacitance-voltage (ECV) technique, with electrochemical C-V Profiler WEP-CVP 21 (WEP, Furtwangen, Germany). The ECV technique is widely applied to measure the majority carriers [10].

In order to reduce the unwanted doping in IV (III-V)-based semiconductors, after the last III-V (IV) deposition, we have applied the growth procedure of depositing on the graphite elements of the MOVPE growth chamber several “coating runs” constituted of IV (III-V) elements materials. An average of 1 µm deposition for each coating run has to be considered. After the coating run depositions, the layer under investigation has been grown, the morphology analyzed, the related growth rate and dopant incorporation measured. We have also considered the possibility to replace some parts of the MOVPE growth chamber over which the deposition takes place. In particular, we have replaced the susceptor, the graphite satellites, the ceiling and the quartz plate (see Figure 1). The graphite ring, usually operating at lower temperature than the other growth chamber parts, has not been replaced.

## 3. Results and Discussion

### 3.1. Ge Deposition

The MOVPE growth of the solar cell structures is usually carried out in mass transport regime, at temperature values around 600 °C, however, in the attempt to minimize the evaporation of the contaminants from the graphite parts of the growth chamber, the growth temperature for the Ge deposition has been reduced to 475–500 °C.

The drawback of reducing the growth temperature is the change of the growth regime: around 470 °C the Ge deposition with GeH_4_ takes place in the kinetic regime [11], therefore it is strongly influenced by temperature variation along the wafer radius, as well as by the total carrier gas flow. In Table 1, we report the growth rate of Ge deposited on Ge substrates in a III-V contaminated MOVPE reactor, as a function of several growth parameters, along with the number of IV elements coating runs.

The Ge growth rate related to MOVPE depositions extracted from Table 1, performed with similar germane partial pressure (from 15 Pa to 25 Pa), as a function of the deposition temperature and of the coating runs number is shown in Figure 2.

It can be pointed out that few IV-based material coating runs are enough to obtain high growth rate, as compared with the values reported in literature (see [11,12]), related to samples grown at similar temperatures and precursor partial pressure. After already one IV elements coating run, Ge growth rate becomes mainly influenced by the MOVPE reactor geometry (which allows obtaining high germane utilization efficiency) and by other growth conditions, like the use of N_2_ as carrier gas instead of H_2_, which improves the growth rate [9].

At this stage, the “As growth blocking role” seems irrelevant, as the growth rate can be easily controlled and optimized, for example, by changing the total flow and the precursors partial pressure, as shown by comparing the runs S3, S4 and S5. The evolution of the background carrier concentration related to the samples S1, S2, S4, S7, S8 and S10, nominally undoped, has been measured by electrochemical capacitance-voltage measurements (see Figure 3a).

In sample S1, a carrier concentration around 10^20^ cm^−3^, (likely due to the incorporation of arsenic, see SIMS analysis reported in Figure 7) has been measured at the beginning of the run. This value is reduced to 1–2 × 10^18^ cm^−3^ after 2 µm thick Ge deposition. During Ge growth, the level of contamination is reduced, as germanium is deposited, with different efficiency, on all the growth chamber surfaces (susceptor and ceiling, etc) from which the previous deposited arsenic can evaporate. From sample S1 to sample S4, the carrier concentration drops at the end of the run from 2 × 10^18^ cm^−3^ to 3 × 10^17^ cm^−3^.

After replacing the susceptor, ceiling, quarts plate and graphite satellites, a significant arsenic contamination is still present at the beginning of run S7, probably due to the residual evaporation of arsenic from the graphite ring, however, this contamination is reduced to 4 × 10^16^ cm^−3^ after 3.5 µm thick Ge deposition. The contamination further decreases to 1 × 10^16^ cm^−3^ in run S8, after a Ge deposition 7.5–8 µm thick. This carrier background value has remained constant even after several further coating runs, as shown by looking at the carrier profile measured in the sample S10 (performed after further 21 coating runs of Ge and SiGeSn layers).

It is worthwhile to point out that the background “contamination” is also related to the growth rate, as shown in Figure 3b. The first part of S9 has been carried out at high growth rate (117 nm/min), while the second part, with a growth rate one order of magnitude lower (10 nm/min). The background carrier concentration increases from 6 × 10^15^ cm^−3^ to 4.5 × 10^16^ cm^−3^ by switching from high to low growth rate. This result can be explained by considering that by increasing the growth rate, the contaminants which are present in the MOVPE growth chamber become much more diluted during the deposition of the material. The drop in carrier concentration registered near the surface of sample S9 could be a measurement artefact or it could be explained by the evaporation of the contaminant during the cool down phase of the MOVPE run.

We found that it is possible to increase the growth rate by 6%, without varying the growth temperature (and keeping constant the total carrier flow and the germane partial pressure), by replacing germane diluted in hydrogen with germane diluted in nitrogen (compare the growth rate values reported in Table 1 for samples S5 and S10).

Remarkably, we have been able to switch Ge conductivity polarity from n-type to p-type as shown in Figure 4. The first part of sample S11 has been grown nominally undoped, then, by injecting in the growth chamber DEZn, with a molar fraction of 3.5 × 10^−4^, the last part of the sample shows a p–type conductivity, with a carrier concentration around 2–3 × 10^15^ cm^−3^. On sample S12, we have replaced DEZn with TMGa and the dopant molar fraction has been increased to 1.8 × 10^−3^. Eventually, sample S13 has been deposited with a constant TMGa molar fraction of 1.27 × 10^−2^. The related ECV characterization shows a constant p-type carrier profile, with a concentration around 1 × 10^18^ cm^−3^. It is worth noting that the doping level of sample S13 is lower than the peak doping level of sample S12, despite a higher input TMGa molar fraction has been utilized for the former sample. This apparent contradiction is due to the higher growth temperature utilized in the deposition of sample S13 (25 °C higher than the one set for the growth of sample S12, see Table 1). The TMGa molar fraction decreases as the temperature increases, because GeH_4_ activation energy is higher than TMGa activation energy (41–42 kcal/mole versus 35.4 kcal/mole, respectively). This means that the effective TMGa molar fraction was actually lower in the sample S13 than in the sample S12.

The carrier concentration results related to runs S11, S12 and S13 show that to a large extent it is possible to modulate the p-type doping in Ge even when the IV element semiconductor is deposited in a reactor which is also utilized for III-V growth.

### 3.2. SiGe(Sn) Deposition

SiGe(Sn) depositions, from sample S14 to sample S17, have also been carried out in a III-V contaminated reactor and after several IV-based coating runs as reported in Table 2.

The growth rate of run S14 (SiGe), coherently with the results presented in the previous chapter, is comparable with that obtained on Ge epitaxial samples, even if a higher GeH_4_ partial pressure has been utilized. It is worthwhile to point out that an increment in GeH_4_ partial pressure has been required to compensate the growth rate decrease caused by the injection of Si_2_H_6_ in the growth chamber (see Figure 5).

Si_2_H_6_ decomposition, in fact, introduces hydrogen atoms that passivate Ge surface, as found for silicon epitaxy [13].

A carrier concentration peak value as high as 4 × 10^19^ cm^−3^ has been measured by EVC on sample S14, deposited after 33 coatings runs (see Figure 6). This value is four times higher than the value measured on sample S4, which was deposited only after five coating runs (see Figure 3a). In order to understand the reason behind this result, we have analyzed the IV elements coating runs performed till the sample S14.

We came to the conclusion that the background carrier concentration peak value measured in sample S14 could be correlated to the three SiGeSn coating runs sequentially deposited just before the run S14. During SiGeSn growth, in fact, HCl can be formed in the growth chamber by the decomposition products of SnCl_4_ and hydrides (Si_2_H_6_ and GeH_4_). Since HCl is recognized as an etchant for Ge and SiGe [14], during SiGeSn deposition, Ge and SiGe coatings already deposited on the reactor graphite parts can be subjected to chemical etching and, as a consequence, possible evaporation paths for the previous deposited As can be generated. The presence of a high arsenic concentration in the sample S14 and in the subsequent samples (S15 and S16) has been assessed by SIMS (see Figure 7a).

Indeed, in all samples a high As incorporation is measured at the beginning of the MOVPE deposition. Since SiGe growth rate is one order of magnitude higher than the SiGeSn growth rate (see Table 2), the As contaminant can be highly diluted in SiGe and its concentration strongly decreases during the SiGe growth, while it remains almost constant in the case of SiGeSn growth. The effect of the growth rate on As incorporation is also evidenced on sample S16, composed of a first Ge buffer layer and of a subsequent SiGeSn layer. For this sample, owing to the high thickness, we joined the results of SIMS and ECV measurements in order to show both As and the carrier profile along the sample depth (see Figure 7a). Like SiGe, Ge deposition takes place at high growth rate, therefore the ECV measurement shows a rapid decrease in the carrier concentration during the deposition. On the contrary, on SiGeSn, SIMS measurement shows that As concentration rises to values as high as 10^19^ cm^−3^.

All these data lead to the following conclusions: (i) in a III-V contaminated MOVPE growth chamber, during SiGeSn deposition a considerable amount of arsenic can be released from the MOVPE reactor walls and be incorporated in SiGeSn and subsequent layers, (ii) the As contamination can be reduced during the deposition by growing the group IV compounds at high growth rate, around 100 nm/min, while it remains at level around 10^19^ cm^−3^ when the growth rate is one order of magnitude lower (10 nm/min).

At this point, it becomes interesting to analyze the reason of the growth rate reduction from run S14 (SiGe) to run S15 (SiGeSn), despite both MOVPE runs have been carried out with the same GeH_4_ partial pressure. Neither the higher Si_2_H_6_ partial pressure utilized in run S15 with respect to run S14, nor the “As growth blocking role” can explain the reduction in growth rate. In fact, referring to the data shown in Figure 5, an increase of Si_2_H_6_ partial pressure from 2.5 Pa to 6.25 Pa can only explain a 20% decrease in the growth rate. Furthermore, the same starting As contamination has been measured in SiGe and SiGeSn (see Figure 7a). This means that As “carry-over” have saturated SiGe and SiGeSn surface sites in the same way. The low growth rate of SiGeSn can still be understood by considering the etching action on Ge and SiGe produced by HCl, as formed by the decomposition products of SnCl_4_, Si_2_H_6_ and GeH_4_. The contrasting actions between the deposition rate and the etching rate has also been observed in GeSn growth by GeH_4_ and SnCl_4_ [15].

In order to reduce the weight of the etching action, a higher germane partial pressure could be used. For example, S. Wirths at al., have shown that it is possible to reach SiGeSn growth rates as high as 125 nm/min with a Ge_2_H_6_ partial pressure of 120 Pa (a value four time higher than the GeH_4_ partial pressure used to grow our samples) [16]. Therefore, in principle, by increasing SiGeSn growth rate it should be possible to reduce the As background concentration near the value obtained in Ge and SiGe.

However, we have also assessed a further difficulty in controlling SiGeSn conductivity, due to the presence of impurities in SnCl_4_ precursor. This is evidenced by SIMS analysis related to P incorporation in the samples S14, S15 and S16 as shown in Figure 7b. It can be pointed out that in run S14 (SiGe) and in the first part of run S16 (Ge growth), where SnCl_4_ is not used, a P concentration, respectively, around 4 × 10^16^ cm^−3^ and 3 × 10^17^ cm^−3^ has been measured, instead, during the growth of SiGeSn (S15), P concentration increases to 3 × 10^19^ cm^−3^.

Thus, we can conclude that by using SnCl_4_, a considerable amount of P is introduced in the growth chamber. Like As, P introduces electrons in IV-based compounds and therefore it influences the material’s conductivity. The P contamination of the growth chamber produced by SnCl_4_ allows explaining the higher P concentration measured in Ge, in run S16, than in SiGe, in run S14. Ge deposition, in fact, has been carried out after SiGeSn growth (S15). During SiGeSn deposition, P is introduced in the growth chamber, can be deposited on the MOVPE reactor walls and then subsequently be released and be incorporated in Ge during the growth. Therefore, for controlling SiGeSn conductivity, regardless the contamination introduced from the previous III-V deposition, it is important to reduce the amount of P contained in SnCl_4_, or to consider alternative precursors, like, for example, TESn (Triethyltin) as, reported for the growth of GeSn [17].

### 3.3. III-V-Based Semiconductor Deposition

We have not observed growth rate variation in III-Vs owing to the previous IV elements growth. Therefore, we have focused our investigation on how to control the III-Vs conductivity, trying to reduce to acceptable levels the IV elements contamination in III-Vs. The III-Vs structures analyzed are reported in Table 3.

MOVPE runs S18 and S19 are related to AlAs/GaAs distributed Bragg reflector (DBR) structures, whose period has been repeated 15 times. These samples have been deposited in mass transport regime (Tgrowth = 888 K) after one and eight III-V coating runs, respectively. In Figure 8, SIMS concentrations of Ge, Si and Sn, measured along the thickness of the samples S18 and S19 are depicted. The incorporation of Ge, Si and Sn is different in AlAs and GaAs, therefore, along the DBR structure thickness, the concentration of these atoms oscillates. After one coating run, the maximum group IV elements concentration in the III-V Bragg test structure is around 5 × 10^17^ cm^−3^. After eight coating runs (with an equivalent deposition of 8 µm thick III-Vs), the maximum background contamination is reduced to values lower than 2 × 10^17^ cm^−3^. This value is more than half of the previously reported value measured in GaAs (4.6 × 10^17^ cm^−3^), in a Si-Ge MOVPE contaminated growth chamber, after a 10 µm thick coating run [18].

In order to further reduce IV elements incorporation in III-Vs, we assessed the influence of the growth temperature on the background carrier concentration. The evaporation of the contaminants from the MOVPE reactor walls is tightly related to the growth temperature and, in particular, by decreasing the growth temperature, we expect reducing the background IV elements concentration in III-V semiconductors.

We have then investigated a InGaP/InGaAs/InGaP/Ge(GaAs) structure (S20), in which the bottom InGaP and InGaAs layers have been deposited at 903 K, while InGaP top layer has been deposited at lower temperature (773 K). Moreover, in run S21 we have replicated the same growth process of run S20 concerning the InGaP and InGaAs bottom layers, while the InGaP top layer has been replaced by the InGaAs one. For both samples, the IV elements concentration has been indirectly measured by ECV, as shown in Figure 9. It is worthwhile to point out that the MOVPE deposition of sample S20 and S21 have been carried out after 20 and 32 III-V coating runs, respectively.

On sample S20, a starting background carrier concentration near 1 × 10^18^ cm^−3^ has been measured when the material is grown at 903 K on Ge substrate. After depositing 1 µm of InGaP at 773 K, the background carrier concentration is reduced to 6 × 10^15^ cm^−3^, a value around three order of magnitude lower than the value measured by E. Welser, in InGaP layers grown in a Ge-contaminated MOVPE growth chamber [8]. This comparison confirms the key role of the growth temperature for decreasing the incorporation of IV elements contaminants in III-Vs. By growing InGaP on a GaAs substrate, the carrier concentration can be further reduced by an order of magnitude, down to 3 × 10^14^ cm^−3^. By using a GaAs substrate, in fact, the Ge auto-doping and Ge solid state diffusion that take place during the InGaP growth on a Ge substrate can be both suppressed [19].

A strong decrease in the background carrier concentration owing to the reduction of the growth temperature has also been measured on sample S21 (see Figure 9b). During InGaP nucleation, which takes place at 903 K, the starting carrier concentration level is as high as 2 × 10^18^ cm^−3^. During the InGaAs deposition carried out with a growth rate twenty-four times higher than the InGaP one, the level of contamination is reduced to 2 × 10^17^ cm^−3^. As soon as the growth is stopped, before the temperature ramping, there is an accumulation of contaminants on the sample surface. Eventually, after the growth of 0.4 µm InGaAs at 773 K, the carrier concentration sharply decreases to 6 × 10^16^ cm^−3^. This value is still higher than the value measured in the top InGaP layer of sample S20, because the top InGaAs layer of sample S21 has been deposited with a thickness which is more than half of the InGaP one.

It is interesting to compare the background carrier concentration measured at the end of InGaAs layer grown at higher temperature on the sample S21 with the IV elements atoms concentration measured by SIMS on sample S19 (see Figure 9b and Figure 8). These values are almost equal, showing that after 8 µm thick III-V coating deposition, the subsequent III-V coatings are not effective in further reducing the back ground carrier concentration.

### 3.4. Morphological and Structural Assessment

The effect of the cross influence between group IV and III-V elements on the morphology and crystal structure of III-V and IV-based semiconductors has been assessed. As shown in Figure 10, Ge surface morphology does not degrade when Ge is grown in a III-V contaminated MOVPE reactor. The surface morphologies of Ge layers are excellent and the best ones have been obtained when Ge is doped with Gallium (sample S12). Similar morphological results have been obtained for SiGe (run S14).

SiGeSn morphology, on the other hands, has been found strongly dependent on the level of As contamination in the MOVPE growth chamber. If SiGeSn is grown by MOVPE, around 756 K, in an arsenic free growth chamber (i.e., in a growth chamber with clean graphite parts that did not see any previous III-V deposition), tin precipitation takes place and the layer’s morphology deteriorates (see Figure 11). In order to explain the key role of As to avoid tin precipitation, a theory on the role of the bond length of adatoms in inhibiting tin segregation has been proposed [20].

Concerning III-Vs semiconductors, we found that when they are deposited in a IV elements contaminated MOVPE growth chamber their morphology changes as a function of the substrate type and orientation. On sample S20, for example, the top InGaP layer, deposited at low temperature, presents a featureless morphology on GaAs substrate and different oriented defects on Ge substrate (see Figure 12).

In particular, regardless of the growth temperature, we assessed that III-V’s morphology on Ge substrate strongly degrades in Sn-contaminated MOVPE growth chamber, eventually leading to a polycrystalline material.

In Figure 13, for example, we show the morphology and the structural characterization of a AlGaAs/GaAs/AlGaAs heterostructure, deposited on Ge and GaAs substrates in a Sn-contaminated MOVPE growth chamber.

The results reported in Figure 13 could be explained by assuming that during the heating phase of the MOVPE run, Sn evaporates from the reactor walls and can be adsorbed on the substrate surface at different concentrations according to the substrate orientation. In particular, the (100) 6° off towards <111> orientation has more steps and kinks, where Sn can be adsorbed, than (100), 2° off towards <110> orientation. If Sn adsorbed somehow disturbs the atoms incorporation in the crystal, the epitaxial growth could be prejudiced on substrates whose orientation favors Sn adsorption. In order to confirm this hypothesis and exclude the influence of the chemical nature of the substrate, further experiments will be reported in a next publication by considering the deposition of III-Vs on Ge substrates with the same orientation of the GaAs ones.

## 4. Conclusions

With this study we have demonstrated that by an adopting an innovative design of the MOVPE reactor and adopting proper growth procedures, it is possible to drastically reduce the cross influence between groups IV and III-V elements when they are deposited in the same MOVPE growth chamber.

We have firstly applied the growth procedure of depositing on the graphite elements of the MOVPE growth chamber several “coating runs” constituted of IV (III-V) elements materials. After the coating run depositions, the layer under investigation has been grown and the related growth rate and dopant incorporation measured. We have demonstrated that few IV-based coating runs are enough to control As “carry-over”, so that the growth of IV-based materials can be carried out without growth rate penalization.

Along with the coating depositions, the temperature and the growth rate are shown to be key parameters for reducing the cross-doping, since the former reduces the evaporation of the impurities from the reactor walls, while the second allows increasing the dilution of the contaminants that are present in the MOVPE growth chamber during material deposition.

In particular, after 8 µm thick deposition of Ge, at growth rate ≥100 nm/min, by decreasing the growth temperature to 720–750 K, the background carrier concentration in Ge has been reduced from 10^20^ cm^−3^ to 3 × 10^17^ cm^−3^. The replacement of some graphite parts of the MOVPE growth chamber (like susceptor, ceiling, quarts plate) has also made it possible to further reduce the background carrier concentration to 6 × 10^15^ cm^−3^. To a large extent, it has then been possible to modulate both the n-type and the p-type conductivities in Ge: n-type and p-type doping has been varied in the range 6 × 10^15^ cm^−3^–10^20^ cm^−3^ and 2–3 × 10^15^ cm^−3^–5 × 10^18^ cm^−3^, respectively.

Even if these results have been demonstrated on Ge, they can be extended to SiGe, since this semiconductor can be deposited at the same growth rate and temperature of Ge.

On the other hand, we found that SiGeSn growth is limited by the etching action produced by HCl, as formed by the decomposition products of SnCl_4_, Si_2_H_6_ and GeH_4_. We have shown that the etching action produced by HCl produces two negative effects that do not allow reducing the group V elements contamination: i) it decreases the SiGeSn growth rate (one order of magnitude with respect to SiGe grown at the same GeH_4_ partial pressure), and ii) it removes the Ge and SiGe coatings already deposited on the reactor graphite parts, thus opening possible evaporation paths for the previous deposited As. Furthermore, we found that SnCl_4_ precursor utilized in the SiGeSn growth transfers a considerable amount of P in the reactor chamber; therefore, regardless of the contamination due to the previous III-V deposition, a reduction of the amount of P impurities in the SnCl_4_ source has to be addressed in order to reduce the background carrier concentration to an acceptable level.

We have also deposited AlAs, GaAs, InGaP and InGaAs in the same MOVPE growth chamber utilized for the growth of Ge, SiGe and SiGeSn. III-Vs semiconductors have been deposited with a growth rate between 9 nm/min and 200 nm/min and in the temperature range 773–903 K. After 8 µm thick III-V based coating, the IV elements contamination has been reduced in AlAs/GaAs Bragg test structures from values higher than 10^18^ cm^−3^ to values around 2 × 10^17^cm^−3^. EVC analysis on InGaP and InGaAs samples has shown that a strong decrease in the background carrier concentration can be further reached by growing the III-V samples at 773 K. Background carrier concentration values as low as 6 × 10^15^ cm^−3^ and 3 × 10^14^ cm^−3^ have been measured in InGaP, respectively grown on Ge and GaAs substrate. These background carrier concentration values are acceptable to manufacture high efficiency solar cells, since the lower p-n junction doping level needed for InGaP solar cells is in the range 1 × 10^16^ cm^−3^–5 × 10^17^ cm^−3^ [21].

Finally, we have assessed the cross influence between group IV and III-V elements on the morphology and crystal structures of III-V and IV-based semiconductors. Ge and SiGe morphologies do not degrade when the IV elements semiconductors are grown in a III-V contaminated MOVPE reactor. SiGeSn morphology, on the other hand, in the temperature range 748–756 K, has been found As-contamination dependent, and, in particular, we discovered that the As contamination present in the MOVPE growth chamber is helpful to avoid the morphology degradation owing to tin precipitation.

The morphology of III-Vs semiconductor when deposited in a IV elements contaminated MOVPE reactor has been found featureless when the deposition takes place on GaAs substrates with orientation (100), 2° off towards <110>. On other hands, the morphology gets worse, in particular in Sn-contaminated MOVPE growth chamber, eventually leading to polycrystalline materials, when the deposition takes place on Ge substrate with orientation 6° off towards <111>.

As a final remark, by considering that the “coating runs” can be replaced by proper buffer layers and by assuming to grow the active parts of III-IV-V Multi-junction (MJ) devices at temperature around 750K–770K, the results reported in this study show that the realization of III-V and IV-based monolithic architectures in the same MOVPE growth apparatus can be feasible for realizing high efficiency MJ solar cells at lower cost.

## Figures and Tables

**Figure 1 materials-14-01066-f001:**
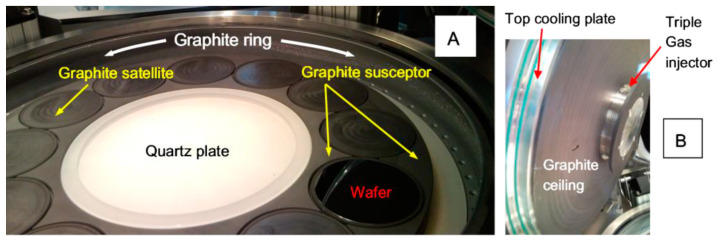
MOVPE growth chamber: (**A**) bottom part; (**B**) top part.

**Figure 2 materials-14-01066-f002:**
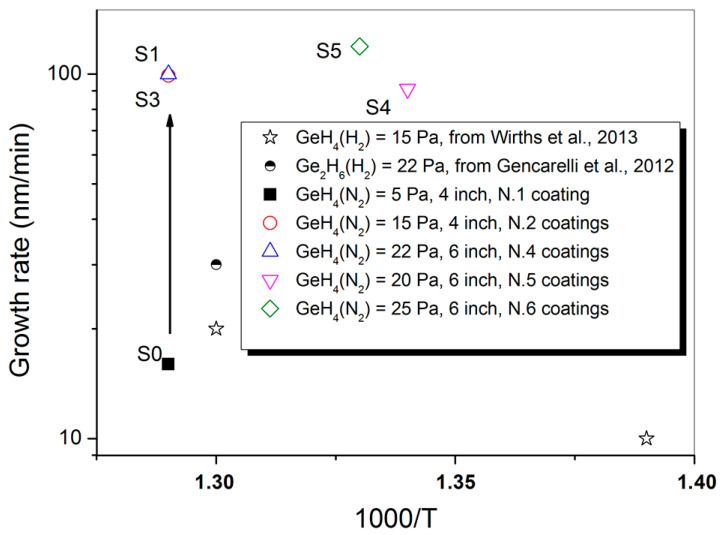
Comparison between Ge growth rate data obtained in this work, by using GeH_4_ and N_2_ as carrier gas, in a III-V contaminated MOVPE reactor (see Table 1) and those reported in Wirths et al., 2013 (ref [11]) and Gencarelli et al., 2012 (ref [12]), where Ge has been grown respectively by GeH_4_ and by Ge_2_H_6_, with H_2_ as carrier gas.

**Figure 3 materials-14-01066-f003:**
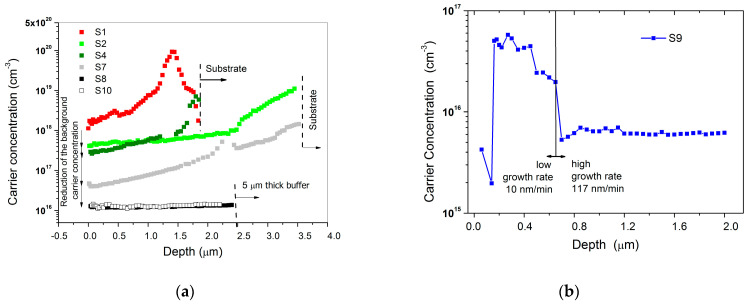
(**a**) evolution of N-type carrier concentration in Ge samples grown in a III-V contaminated MOVPE reactor after different Ge and SiGeSn coating deposition; (**b**) variation of the N-type carrier concentration in sample S9 as a function of the growth rate. Measurements performed by ECV technique.

**Figure 4 materials-14-01066-f004:**
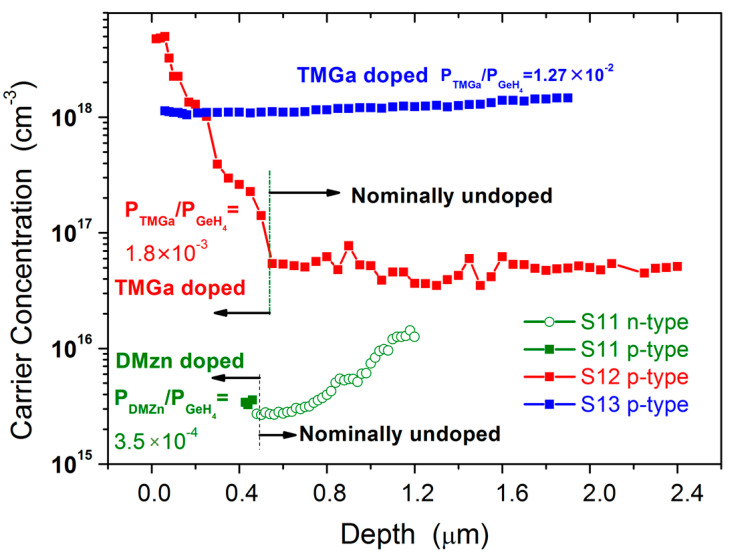
N-type and p-type carrier concentration in Ge samples S11, S12 and S13 measured by ECV. The first part of sample S11 (S12) is grown nominally undoped, the last part of the layer is grown by injecting in the MOVPE growth chamber a DMZn (TMGa) molar fraction equal to 3.5 × 10^−4^ (1.8 × 10^−3^). Sample S13 has been grown by injecting a constant TMGa molar fraction equal to 1.27 × 10^−2^.

**Figure 5 materials-14-01066-f005:**
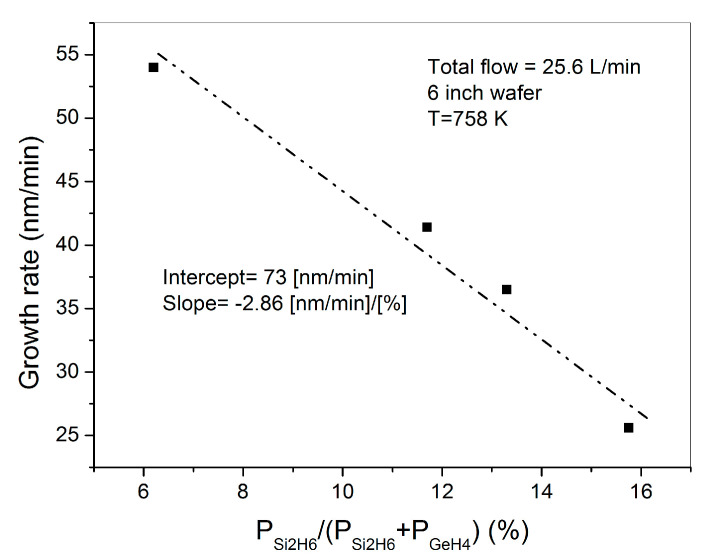
SiGe growth rate as a function of Si_2_H_6_ composition in the gas phase.

**Figure 6 materials-14-01066-f006:**
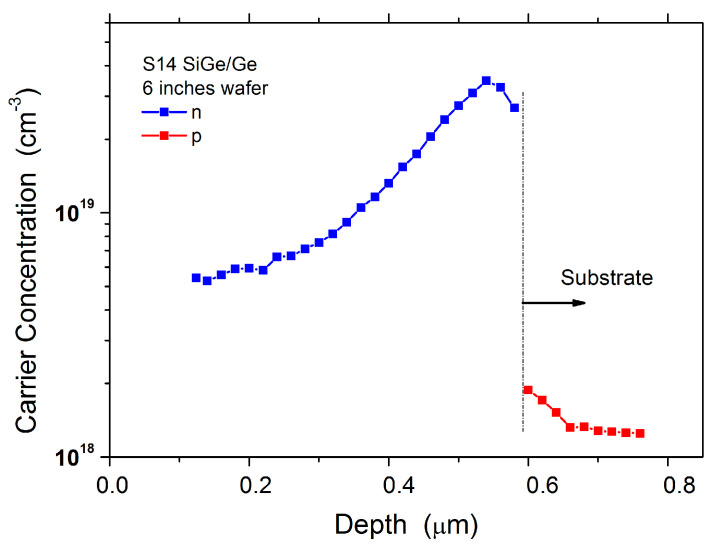
ECV carrier profile measured on the sample S14.

**Figure 7 materials-14-01066-f007:**
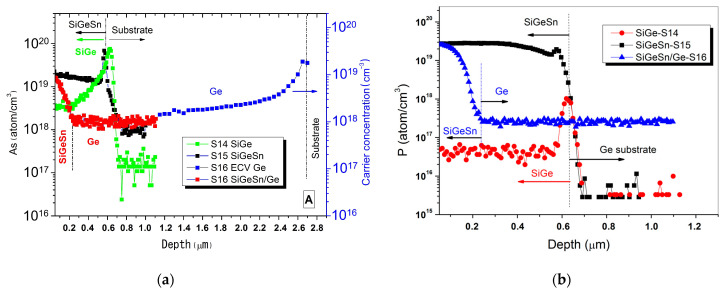
SIMS profile of As (**a**) and P (**b**) in samples S14, S15 and S16 (see Table 2). The As profile of sample S15 has been stopped around 1.1 µm depth. Subsequently, the carrier concentration profile has been measured by ECV.

**Figure 8 materials-14-01066-f008:**
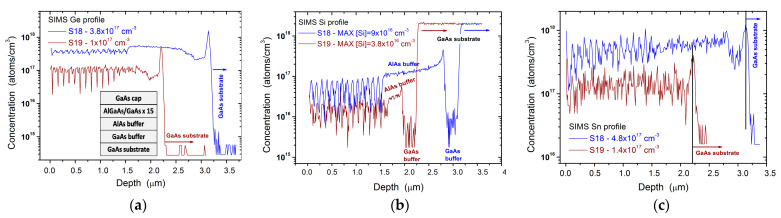
SIMS profile of: (**a**) Ge, (**b**) Si and (**c**) Sn in samples S18 (deposited after one III-V coating run) and S19 (deposited after eight III-V coating runs). In the inset, the DBR structure is shown.

**Figure 9 materials-14-01066-f009:**
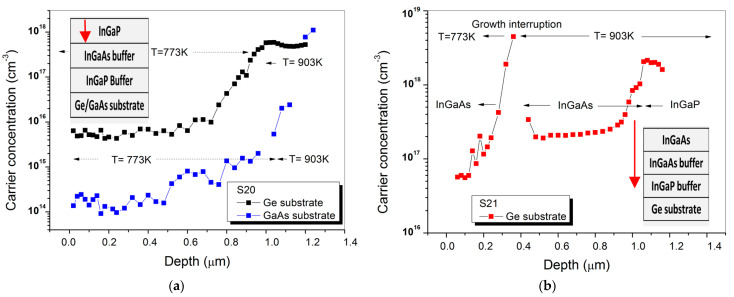
ECV Profile on: (**a**) sample S20 and (**b**) on sample S21. In the insets, the structures investigated are depicted. The arrows indicate the etching depth.

**Figure 10 materials-14-01066-f010:**
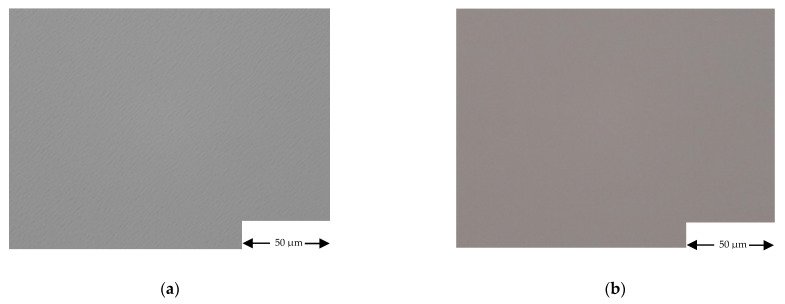
Optical microscope characterization of samples: (**a**) S6 and (**b**) S12. Magnification 500×.

**Figure 11 materials-14-01066-f011:**
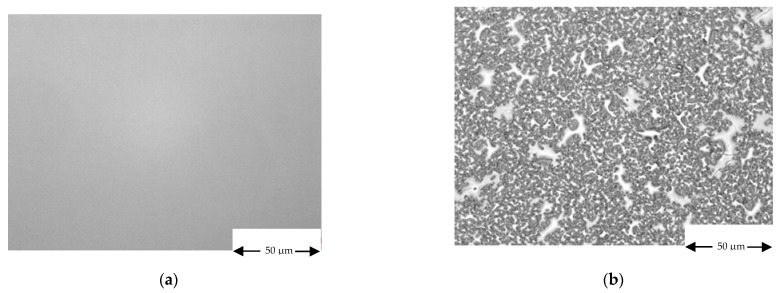
SiGeSn morphology in: (**a**) As contaminated and (**b**) in a As free MOVPE growth camber, related, respectively, to the samples S15 and S17. The white islands depicted on sample S17 are related to tin precipitation.

**Figure 12 materials-14-01066-f012:**
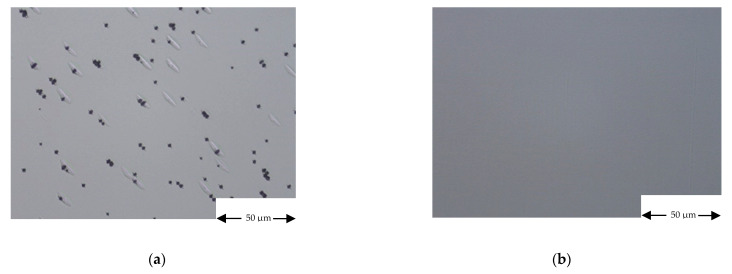
Run S20 InGaP morphology on (**a**) on Ge substrate and (**b**) on GaAs substrate. Magnification 500×.

**Figure 13 materials-14-01066-f013:**
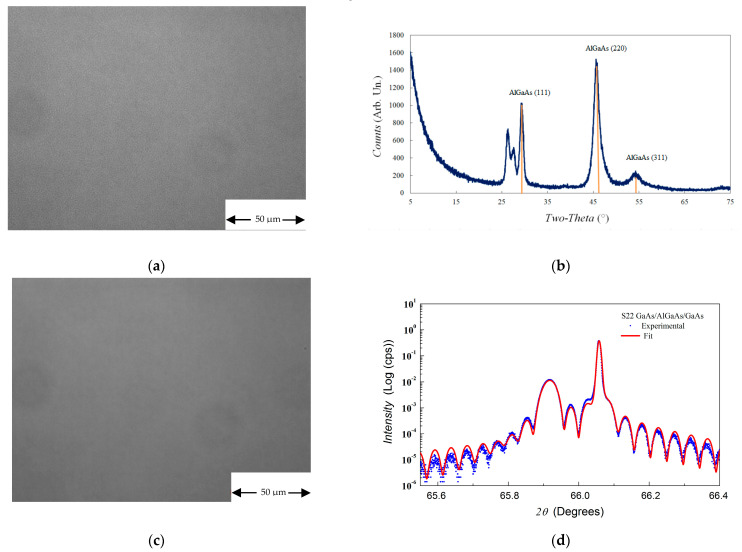
GaAs/AlGaAs/GaAs structure morphology and XRD (GIXRD) characterization. The structure has been grown in a Sn-contaminated MOVPE growth chamber. (**a**) morphology on Ge substrate, (**b**) GIXRD characterization on Ge substrate, showing the polycrystalline nature of the deposited layers, (**c**) morphology on GaAs substrate, (**d**) XRD characterization on GaAs substrate, the intense interference fringes are an index of the sharpness of the interfaces i.e., of their low roughness.

**Table 1 materials-14-01066-t001:** Ge Growth rate obtained on 4- and 6-inch wafers as a function of growth parameters and of the number of IV-based materials coating runs.

N Run	T Growth (K)	Total Flow (L/min)	GeH_4_ Partial Pressure (Pa)	Wafer Diameter (inch)	N. of Coating Runs of IV-Based Materials after the Last III-V Growth	Growth Rate in the Wafer Center (nm/min)
**S0**	773	20.5	5.5	4	1	16
**S1**	773	21	15	4	2	99
**S2**	773	20.7	61	6	3	111
**S3**	773	20.5	22	6	4	100
**S4**	745	22	20	6	5	91
**S5**	750	18	25	6	6	119
**S6**	773	21.6	21	6	9	100
**Replacement of Growth Chamber Reactor Parts**						
**S7**	748	18	25	4	1	-
**S8**	748	18	25	4	12	-
**S9**	748	18	25/2.2	4	13	117
**S10**	753	18	25	4	21	126 (*)
**S11**	723	18	25	4	22	(*)
**S12**	723	18	25	4	24	(*)
**S13**	748	18	24	4	25	(*)

Note: All the runs have been performed with nitrogen as carrier gas and germane diluted in hydrogen, if not otherwise stated. The growth rate has been measured in the center of the samples. (*) GeH_4_ diluted in N_2_.

**Table 2 materials-14-01066-t002:** SiGe(Sn) growth rate obtained on Ge wafers as a function of the growth parameters and of the IV elements coating runs number.

N Run	T Growth (K)	Total Flow (L/min)	GeH_4_ Partial Pressure (Pa)	Si_2_H_6_ Partial Pressure (Pa)	SnCl_4_ Partial Pressure (Pa)	N. of Coating Runs of IV-Based Materials after the Last III-V Growth	Growth Rate (nm/min)
S14 (SiGe)	748	12	37.9	2.5	NA	33	100
S15 (SiGeSn)	756	12	37.9	6.25	1.34	39	7
S16 (SiGeSn/Ge)	756	12/18	37.9/ 37.9	6.25/0	1.34/0	43	-
Replacement of growth chamber reactor parts							
S17 (SiGeSn)	756	12	37.9	8.12	1.7	1	-

Note: All the runs have been performed with nitrogen as carrier gas and germane diluted in hydrogen, if not otherwise stated. The growth rate has been measured in the center of the samples.

**Table 3 materials-14-01066-t003:** Growth condition related to III-Vs deposition in a MOVPE growth chamber contaminated by IV elements.

N Run	T Growth (K)	Total Flow (L/min)	Growth Rate (nm/min)	N. of Coating Runs of III-Vs after the Last IV-Based Growth	Substrate
S18 (AlAs/GaAs DBR)	888	12	40/40	1	GaAs
S19 (AlAs/GaAs DBR)	888	12	40/40	8	GaAs
S20 (InGaP/InGaAs/ InGaP)	773/903/903	12	11/200/9.1	20	Ge and GaAs
S21 (InGaAs/InGaAs/ InGaP)	773/903/903	12	66/200/9.1	31	Ge
S22 (GaAs/AGaAs/ GaAs)	764	12	-/28.3/-	1 (*)	Ge and GaAs

Note: (*) after SiGeSn deposition.

## Data Availability

The data presented in this study are available on request from the corresponding author.

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
