# Peer review of "Study of the Cross-Influence between III-V and IV Elements Deposited in the Same MOVPE Growth Chamber"

_materials, 2021, doi:10.3390/ma14051066_

Round 1

Reviewer 1 Report

The authors have presented a manuscript with the title of “Study of the cross-influence between III-V and IV elements deposited in the same MOVPE growth chamber”. 

In this work, the authors proposed effective methods like using a modified MOVPE growth chamber, depositing III-V (IV) elements “coating runs”, and optimize the growth temperature and growth rate etc, to make the integration of group IV with III-V compounds in the same MOVPE growth chamber feasible.

The manuscript is very well written. The experimental and characterization work are solid. This work may be interested to the researchers/readers in the related community. I recommend the publication of this manuscript (pending possible minor revision).

There are some typo:
1.    Line 357 on page 11, it seems there is a display issue of the sentence.
2.    Line 421 on page 14, 3*10^17 (there is a typo);

Overall, this is quite a good work. The reviewer is happy to recommend publication this work.

Author Response

  1. Line 357 on page 11, it seems there is a display issue of the sentence. Correction introduced
  2. Line 421 on page 14, 3*10^17 (there is a typo);  Correction introduced

Author Response

Page 6, Lines 187: The abbreviations DEZn and TMGa are unclear. A short explanation needs to be added to the experimental section. Chemical formulas have been added  in the section 2 Materials and methods

Page 9, Line 270: [15]x. Correction introduced

Page 10, Figure 8: The quality of Figure 8 is very low, for example the labels on AlAs/GaAs DBR structure is unreadable. And the same about Figure 9.

New structures have been introduced in Fig 8 and 9 replacing the unreadable images

Page 11. Line 357: Errore. L'origine riferimento non è stata trovata  Correction introduced

Reviewer 3 Report

The topic of the manuscript is live, the work is well focused and overall clearly presented. However, some changes would be required before publication.

-The abstract summarizes and intriguingly enumerates the optimal conditions... it is kind of difficult to read. The authors may want to improve this aspect. The same applies to the conclusions.

-Could you provide more details (technical) about your custom made species injector?

-Please indicate vendors of the equipment used (also for characterization).

-Please provide more details on " the growth procedure of depositing on the graphite elements of the MOVPE growth chamber several "coating runs" 

-Please provide a reference for: "The drawback of reducing the growth temperature is the change of the growth regime: around 470C the Ge deposition with GeHl4 takes place in the kinetic regime and it is strongly influenced by temperature variation along the wafer radius, as well as by the total carrier gas flow".

-Please provide a specific name and reference of the technique in "has been measured by electrochemical capacitance-voltage measurements (see Figure 3 (a))."

-Pag 4, 148: use the proper symbol for micron

-There is an extra "x" in line 270

-Please fix the broken link in line 357. 

-The last part of the manuscript remains open: " In order to confirm this hypothesis and exclude the influence of the chemical nature of the substrate, further experiments should be carried out by considering the deposition of lll-Vs on Ge substrates with the same orientation of the GaAs ones" . Actually, this is an important point; potentials readers would be very interested in those experiments and those would support your manuscript and hypothesis without leaving the end of the manuscript to more speculative assumptions.

Author Response

-The abstract summarizes and intriguingly enumerates the optimal conditions... it is kind of difficult to read. The authors may want to improve this aspect. The same applies to the conclusions.

Modifications of the abstract and of the conclusion have been introduced

-Could you provide more details (technical) about your custom made species injector?

As indicated in the paper, with respect to the standard triple gas injector  the diameter has been increased. As far as the standard injector is concerned , the reviewer can have a look at the United States Patent

“MOCVD REACTOR HAVING CYLINDRICAL GAS INLET ELEMENT” USOO8841.221B2, Brien et al.  Date of Patent: Sep. 23, 2014

-Please indicate vendors of the equipment used (also for characterization).

In section 2 Materials and methods, the vendors have been indicated for all the equipment used in the experimentation as well as for the characterization tools

-Please provide more details on " the growth procedure of depositing on the graphite elements of the MOVPE growth chamber several "coating runs" 

 The thickness of each coating run has been introduced in the section 2 Material and methods

-Please provide a reference for: "The drawback of reducing the growth temperature is the change of the growth regime: around 470C the Ge deposition with GeHl4 takes place in the kinetic regime and it is strongly influenced by temperature variation along the wafer radius, as well as by the total carrier gas flow".

A reference has been introduced

-Please provide a specific name and reference of the technique in "has been measured by electrochemical capacitance-voltage measurements (see Figure 3 (a))."

In Fig 3 the name of the characterization technique has been included 

-Pag 4, 148: use the proper symbol for micron

Correction introduced

-There is an extra "x" in line 270

Correction introduced

-Please fix the broken link in line 357. 

Correction introduced

-The last part of the manuscript remains open: " In order to confirm this hypothesis and exclude the influence of the chemical nature of the substrate, further experiments should be carried out by considering the deposition of lll-Vs on Ge substrates with the same orientation of the GaAs ones" . Actually, this is an important point; potentials readers would be very interested in those experiments and those would support your manuscript and hypothesis without leaving the end of the manuscript to more speculative assumptions.

We agree with the reviewer that this part remains open. We have ordered new Ge substrates with the same orientation of the GaAs ones, however, before performing this experimentation, we have other important commitments to be completed and we are forced to postpone in a next publication the opportunity to add this piece of information for a final interpretation of the experimental results. We have tentatively included in the text that this issue will tackled in a next publication (see pag.14)

Despite the interpretation of these experimental results remains open, we think it is worthwhile to show to readers that there are critical morphology aspects related to the growth of III-V in a Sn contaminated  MOVPE growth chamber that need to be addressed. However, this part is not the more relevant one of our contribution, so we leave to the reviewer and to the editor the possible decision to remove this section, because the interpretation of the experimental results has still be concluded or to leave it as it is, even if it is incomplete, because the experimental results are anyway judged interesting by themselves.

Round 2

Reviewer 3 Report

Regarding my previous comment: Please provide a specific name and reference of the technique in "has been measured by electrochemical capacitance-voltage measurements (see Figure 3 (a))."

Please inticate what ECV stands for and provide a reference.

Author Response

In section 2 Materials and Methods , the following sentence was added: " the carrier profiles have been extensively measured by electrochemical capacitance-voltage (ECV) technique, with electrochemical C-V Profiler WEP-CVP 21. The ECV technique is widely applied to measure the majority carriers [10]

A reference has been also addeed as suggested.